# Ionic Liquid–Ultrasound-Based Extraction of Biflavonoids from *Selaginella helvetica* and Investigation of Their Antioxidant Activity

**DOI:** 10.3390/molecules23123284

**Published:** 2018-12-11

**Authors:** Yongmei Jiang, Dan Li, Xiankui Ma, Fengqin Jiang, Qun He, Shaoliang Qiu, Yan Li, Gang Wang

**Affiliations:** School of Pharmacy, Zunyi Medical University, Zunyi 563003, China; 18385042506@163.com (Y.J.); 13984515977@163.com (D.L.); 13407141474@163.com (X.M.); 13697793254@163.com (F.J.); hequn111@sohu.com (Q.H.); m18311546781@163.com (S.Q.); 13312429268@163.com (Y.L.)

**Keywords:** *Selaginella helvetica*, ionic liquids, biflavonoid, ultrasound extraction, HPLC, response surface methodology, antioxidant activity

## Abstract

As a new and green solvent, ionic liquids (ILs) have received more attention during the green extraction and separation process for natural medicines. In this paper, IL-ultrasound-assisted extraction (IL-UAE) of total biflavonoids (TBFs) from *Selaginella helvetica* was firstly developed, and different ILs were employed and compared. Based on single-factor experiment, solid–liquid ratio (1:10–1:14 g/mL), IL concentration (0.6–1.0 mmol/mL), and extract temperature (40–60 °C) were further explored, according to response surface methodology (RSM), with TBF yields as the index. Moreover, antioxidant activity of TBF extract was analyzed by four methods, i.e., 2,2-di(4-tert-octylphenyl)-1-picrylhydrazyl (DPPH) and 2,2′-azinobis-(3-ethylbenzth-iazoline-6-sulphonate (ABTS) free radical scavenging assay, ferric ion reducing power assay, and chelation of ferrous ions assay. The results indicated that [C_6_mim]PF_6_ had a high selectivity and efficiency. Moreover, important parameters for the extraction process were investigated and optimized. Through parameter optimization (0.8 mmol/L, 250 W, 40 min, 1:12.7 g/mL, and 47 °C), a yield of 18.69 mg/g biflavonoids was obtained from the extract of *S. helvetica*. Compared with ethanol-UAE, heat-reflux extraction, Soxhlet extraction, and percolation extraction, IL-UAE could not only obtain higher yield in a shorter time, but also reduce the solvent consumption. In addition, TBF extract showed potential antioxidant activity based on the above four antioxidant methods. In short, IL-UAE was first employed to develop a novel and green extraction method for TBF content, and this experiment provides valuable references for further utilization of *S. helvetica*.

## 1. Introduction

*Selaginella helvetica*, is a traditional Chinese medicine which grows in Northeast China and belongs to the *Selaginella* genus [1]. According to traditional Chinese medical theories, it was beneficial for reducing fever, removing toxicity, activating blood circulation, removing blood stasis, relieving cough, reducing sputum [2], and more. The latest studies had reported that it was mainly used for the treatment of laryngopharyngeal swelling, rheumatism, nasopharyngeal carcinoma, asthma, and traumatic bleeding [3]. The major chemical constituents of *S. helvetica* include biflavonoids, alkaloids, sterols, and organic acids [4].

Biflavonoids consist of two mono flavonoids with complex and diverse properties [5]. Since Okigawa etc. [6] first separated five biflavonoids in 1971 from *S. tamariscina*, scholars have separated more than 30 biflavonoids. Amentoflavone, ginkgetin, hinokiflavone and heveaflavone (see Figure 1), known as biflavonoids, are active phytochemicals derived from *S. moellendorffii*, *S. tamariscina*, and *S. delicatula* [7]. Most biflavonoids have prominent antioxidant characteristics [8,9], and have a broad range of pharmacologic activity, such as anticancer, antiviral, anti-inflammatory, anti-Alzheimer’s disease, anti-myocardial ischemia, and antimicrobial effects [10,11].

Meanwhile, Li and colleagues [12] considered that the formation of some diseases, such as occult malignancy, senescence, myocardial ischemia, cerebral ischemia, atherosclerosis, Alzheimer’s disease, and Parkinson’s disease, had important relations with oxidative damage in the human body. The reason was that reactive oxygen species (ROS) were produced in the body, which contributes to an array of normal physiology metabolism, and clears excess free radicals at the same time under normal conditions. However, if the reaction mechanisms are damaged, the excess free radicals will affect the human body, leading to the damage of tissues and cells. Therefore, when the free radicals were excessive or human antioxidants and repair function was damaged, it can cause oxidative stress damage, producing different diseases [13]. It has always been a research hotspot to explore natural antioxidants in plants [14]. There are some reports that flavonoids, phenolic acid, and coumarins can be used as potential antioxidants [15]. However, the biflavonoids of *S. helvetica* had not been focused on during the antioxidant research.

Ionic liquid (IL) is a molten salt which consists of different cations and anions at room temperature. In recent years, ILs have also received more attention as a green solvent because of their particular characteristics, including high polarity, low viscosity, good hydrophobicity, and accurate selectivity [16]. Nowadays, IL has been widely applied in the extraction and separation process for natural medicine, such as flavonoids [17], phenolic acids [18], alkaloids [19], terpenoids [20], and other compounds. Most importantly, IL can be designed by various combinations of different anions and cations [21] to change its physical and chemical properties and can infiltrate quickly into the plant cell and dissolve the targets.

Recently IL-ultrasound-assisted extraction (IL-UAE) has been found for natural product extraction and has many advantages, including less organic solvent usage, shortening of extraction time, high extraction efficiency, and is also environmentally friendly [22]. For instance, Xu et al. [23] extracted curcuminoid constituents from rhizomes of *Curcuma longa* with IL-UAE. First of all, after screening different ILs, [C_8_min]Br had remarkable extraction efficiency. The response surface methodology (RSM) was adopted to optimize experimental parameters on the curcuminoid yields, and then compared with heat-reflux extraction and ethanol-UAE technique. The results illustrated that IL-UAE can not only enhance the extraction efficiency of several curcuminoids, but also shorten extraction time.

Thus, the aim of this study was first to develop a convenient, effective, and rapid IL-UAE method for the extraction of total biflavonoids (TBF) from *S. helvetica*. After screening the extraction effect of different ILs by various couplings of different anions and cations, the extraction parameters (IL concentration, ultrasonic power, solid–liquid ratio, ultrasonication time, and extraction temperature) were explored and optimized based on a single-factor experiment and RSM design. Several classical extraction methods, including ethanol-UAE, percolation extraction, Soxhlet extraction, and reflux extraction, were compared with the established IL-UAE approach. In addition, antioxidant activities of the TBF extract were investigated for the sake of evaluating its potential value as a natural antioxidant supplement.

## 2. Results and Discussion

### 2.1. Screening Different ILs

#### 2.1.1. Effect of Anion

The screening effect of the IL anion on its extraction efficiency for analytes is considered to be a critical factor, and has significant effects [24,25]. In this study, the 1-butyl-3-methylimidazolium ILs with six different classes of anions (Br^−^, BF_4_^−^, OAC^−^, PF_6_^−^, Cl^−^, NO_3_^−^) were evaluated, and the differences for TBF content were readily evident. As shown in Figure 2A, the yields of TBFs using ionic liquids with PF_6_^−^ and BF_4_^−^ were higher than those of other ILs with the same cation, as well as PF_6_^−^, showing the best results on account of the H-bond generated and the hydrophobic interaction. The strong interaction between PF_6_^−^ with ethanol molecules results in the effective permeation of IL solution into the cells for a high solubility of target products. Moreover, the hydrogen bonding of IL with PF_6_^−^ may generate strong dissolution capacity with the biflavonoids. Thus, PF_6_^−^ was the optimum anion, and chosen for extraction of TBFs from *S. Helvetica*.

#### 2.1.2. Effect of Cation

The alkyl chain length of cation will also affect an IL’s extraction effectiveness towards target compositions. In this study, six ILs with the same anion of PF_6_^−^, and using diverse 1-alkyl-3-methylimidazolium cations, containing C_2_mim^+^, C_4_mim^+^, C_6_mim^+^, C_8_mim^+^, C_10_mim^+^, and C_12_mim^+^, were evaluated on the yields of TBF. Figure 2B showed that [C_6_mim]PF_6_ had a remarkable extraction effect towards target compounds among these six ILs. As the alkyl chain length was increased, the TBF contents were significantly enhanced because of the addition of lipotropism and the hydrophobic effect [26]. Nevertheless, when carbon chain lengths exceeded C-8, TBF yields were clearly decreased. The possible reason is that steric hindrance effects were gradually enhanced between ILs and biflavonoids, and the hydrophobic effect declined little by little. Based on the above results, C_6_mim^+^ was the best cation, and selected for further research.

Considering the extract results of both anion and cation, [C_6_mim]PF_6_ was screened for use in the next experiments.

### 2.2. Single-Factor Experiment

In the initial study, the influence of five factors, including ultrasonic power, solid–liquid ratio, extraction temperature, IL concentration, and ultrasonication time, on the extraction of TBFs from *S. helvetica*, were researched and evaluated. According to the literature of He et al. [27], it is found that imidazoles had the highest solubility in a series of ethyl alcohols at the same temperature, so we used ethanol to dissolve the ionic liquids.

#### 2.2.1. Effect of IL Concentration

The IL concentration can be regarded as the primary factor to be optimized. The extraction effects of different IL concentration (0.2, 0.4, 0.6, 0.8, 1.0, and 1.2 mmol/L) on TBF yields were analyzed in detail. As shown in Figure 3A, when the IL concentration was changed from 0.2 to 0.8 mmol/L, TBF yields were enhanced from 10.63 to 18.17 mg/g. This might be because [C_6_mim]PF_6_ could easily destroy the structure of cell membranes, infiltrated quickly into plant cells and dissolved more biflavonoids, and promoted the addition of TBF content [28]. However, the extraction efficiency was constantly decreased with further increasing of [C_6_mim]PF_6_ concentrations. Seeing that the viscosity of IL solution was added little by little, and its liquidity reduced on a continuous basis, the targets were distinctly influenced and the IL’s extraction efficiency was decreased [29]. Therefore, [C_6_mim]PF_6_ concentrations of 0.6, 0.8, and 1.0 mmol/L were investigated for the subsequent optimization study of RSM.

#### 2.2.2. Effect of Extracting Temperature

Ultrasonic temperature was also an important factor for the biflavonoid extraction. Under the established extraction conditions (l:12 g/mL of solid–liquid ratio, 0.8 mmol/L of IL solution, 250 W of extraction power, 40 min of extract time), the effects of different ultrasonic temperatures (20, 30, 40, 50, 60, and 70 °C) on TBF yields were evaluated in detail. As shown in Figure 3B, with ultrasonic temperature improving continually from 20 °C to 50 °C, the TBF content of *S. helvetica* was obviously increased, changing from 8.54 to 18.19 mg/g. The results indicated that the mass transfer of targets was clearly influenced, and the biflavonoids was quickly dissolved in [C_6_mim]PF_6_ solution to improve TBF content. However, the yields of TBF were distinctly reduced with further increasing extraction temperature. A plausible reason for the low TBF yield could include thermal instability and phenolic oxidation at isolation temperatures >50 °C [30,31]. Therefore, ultrasonic temperatures of 40, 50, and 60 °C would be examined and further optimized in the later experiment of RSM.

#### 2.2.3. Effect of Solid–Liquid Ratio

The ratio of liquid to raw was a crucial influence factor for the extraction process of TBF because of changing the contact area between solids and liquids. The established extraction conditions were set as follows: 50 °C extraction temperature, 0.8 mmol/L of IL solution, 250 W of extraction power, 40 min of ultrasonication time, and the effects of different ultrasonic temperatures (20, 30, 40, 50, 60, and 70 °C) on TBF yields were studied in detail (see Figure 3C). As the solid–liquid ratio was changed from 1:6 to 1:12 mg/mL, TBF yields increased from 8.62 to 18.17 mg/g, which indicated that a large amount of solvent could dissolve more biflavonoids by permeating into plant cells effectively [32]. However, the extraction efficiency was constantly decreased with the further increasing of the ratios of liquid to raw material. The possible reason was that an excess solvent to solid ratio could produce a lot of impurities [33], and cause a significant reduction of TBF content. This indicated that a proper solid–liquid ratio was enough for the extraction. Therefore, solid–liquid ratios of 1:10, 1:12, and 1:14 g/mL were evaluated for the next optimization study of RSM.

#### 2.2.4. Effect of Ultrasonic Power

Under the set extraction process parameters (50 °C of extraction temperature, 0.8 mmol/L of IL solution, l:12 g/mL of solid–liquid ratio, 40 min of ultrasonication time), the effects of different ultrasonic power (150, 200, 250, 300, 350, and 400 W) on TBF yields were evaluated. Figure 3D showed that TBF yields were significantly increased from 13.12 mg/g to 18.22 mg/g with an increase in ultrasonic power from 150 to 250 W. However, when the ultrasonic power was in the range of 250–400 W, TBF content had almost no change. It might be that the method which had mechanical effect, acoustic cavitation, and thermal effect characteristic [34], could increase the velocity of molecular movement and the penetration of the intermediate, thereby increasing the extraction efficiency. Thus, an ultrasonic power of 250 W was used as the optimal factor.

#### 2.2.5. Effect of Extraction Time

The established extraction process parameters were set as follows: 50 °C of extraction temperature, 0.8 mmol/L of IL concentration, l:12 g/mL of solid–liquid ratio, 250 W of extraction power, and the effects of different extraction time (10, 20, 30, 40, 50, and 60 min) on TBF yields were investigated (see Figure 3E). Generally, extraction time can affect the extraction effects to some extent, as less extraction time leads to incomplete extraction in the yields of TBFs. After reaching full extraction, excess time resulted in waste and increased energy consumption. Thus, it was necessary to find a proper condition based on the results of the above screening. Figure 3E showed that the optimum extraction time was 40 min, and the maximum yield of total biflavonoids was 18.24 mg/g. Hence, the ultrasonication time of 40 min was chosen as the optimal factor.

### 2.3. Analysis of Response Surfaces

#### 2.3.1. Fitting the Model

It is well-known that RSM optimization is much better than classic single-factor optimization for natural medicine extraction [35], due to the interaction effects of different variates. For the sake of further investigation of the relationship among the factors in IL-UAE, several independent variables (solid–liquid ratio, IL concentration, extraction temperature) and three levels (0.6, 0.8, 1.0 mmol/L; 1:10, 1:12, 1:14 g/mL; 40, 50, 60 °C) were optimized by Box-Behnken design of RSM on the foundation of the aforementioned single-factor test.

Table 1 exhibited the predicted quadratic model based on ANOVA. The results showed that the recommended experimental models were significant with *p* < 0.0001, the coefficient of determination (*R*^2^) of TBF content was 0.9957, and the adjusted coefficients of determination (Adj. *R*^2^) of TBF was 0.9902. Moreover, the lack of fit of TBF content was not remarkable (*p* > 0.05). The model terms of X_1_, X_2_, X_3_, X_1_X_3_, X_1_^2^, X_2_^2^, and X_3_^2^ were significant according to their *p* values (<0.05). Meanwhile, the influential order of three variables was IL concentration > ultrasound temperature > ratio of liquid to raw material, which demonstrates that IL concentration always plays a significant role in IL-UAE. Therefore, the model was suitable to predict the variation of TBF yield accurately.

The equation showed that enhancing solid–liquid ratio (X_1_), IL concentration (X_2_), and extraction temperature (X_3_) could clearly increase biflavonoid yields (Y). Besides the interactive mode, X_1_X_2_ and X_2_X_3_ had an inverse correlation action, and X_1_X_3_ had a positive relationship action on TBF yields. Between the predicted values and the actual values for TBF, Figure 4 indicated that there was a significant correlation. As introduced by each fact point, the fine fit of both patterns had a tight relation with the corresponding regression line.
Y = 18.37 − 0.71X_1_ − 0.30X_2_ − 1.45X_3_ − 7.5 × 10^−3^ X_1_X_2_ + 0.63X_1_X_3_ − 0.075X_2_X_3_ − 1.76X_1_^2^ − 1.72X_2_^2^ − 1.86X_3_^2^(1)

#### 2.3.2. Effect of Extraction Parameters

The interaction effectiveness was well-exhibited from the 3D RSM between any two variates when another factor always remains on a fixed ideal level. The relationships between dependent variates (total biflavonoid content) and three factors (solid–liquid ratio, IL concentration, and extraction temperature) were illustrated in Figure 5.

The ordinate exhibited the TBF yields and the abscissa exhibited any two variates. The 3D models of RSM could imply how two dependent variables affect the test results. Figure 5 showed that all response surface figures were upward convex with a maximum value of TBF content at the center of the 3D model graph, which indicated the rationalization of the prediction models [36]. The optimum craft parameters were acquired on the basis of quadratic model equation, and were as follows: 0.78 mmol/L IL concentration, 1:12.73 g/mL solid–liquid ratio, and 47.27 °C ultrasonic temperature, giving a predicted TBF yield of 18.6675 mg/g. The calibration tests were slightly adjusted at the optimum factors of 0.8 mmol/L IL concentration, 1:12.7 g/mL ratio of liquid to raw material, and 47 °C ultrasonic temperature. The optimum extraction result of TBF content was found to be 18.69 mg/g by calculating the average of triplicate runs, and was almost the same as the predicted value, which proves that the RSM model was suitable for predicting the expected optimization process.

Furthermore, the morphological characterizations before and after extraction of raw materials using IL-UAE were observed by 8100 scanning electron microscope (SEM, Hitachi High-Technologies Corporation, Tokyo, Japan). As shown in Figure 6, the untreated *S. helvetica* (Figure 6A,A1) clearly showed a thick cell wall, integrated cellular structure, and distinct boundary between different tissues. However, the SEM results of Figure 6B,B1 showed that the surface and cell wall structures of *S. helvetica* were almost destroyed after IL-UAE, which contributed to the dissolution of TBFs into the IL solution. It might be that ultrasonic waves produce violent vibrations and have a very strong destructive function on raw material, which can make cellular tissue deformation, accelerate solvent into the cell wall, and increase the cell wall penetrability in the cell [37].

### 2.4. Recovery of TBF and IL

After extraction, a variety of insoluble organic reagents which were the ideal solvents to recover the targets and IL [38] were investigated for the extraction of TBFs from IL. The recovery of TBFs and IL were calculated according to their concentration ratio before and after extraction using HPLC analysis. As shown in Table 2, the results indicated that only ethyl acetate can obtain good recovery of both TBFs and IL. The vast majority of TBFs (96.23 ± 0.53%) and IL (97.67 ± 0.44%) were recovered from *S. helvetica* extract.

### 2.5. Comparison of Five Extraction Methods

Five methods, including IL-UAE, ethanol-UAE, heat-reflux extraction (HRE), Soxhlet extraction (SE), and percolation extraction (PE), were employed and compared for their biflavonoid yields from *S. helvetica*. As shown in Figure 7, compared with SE, PE, and HRE on TBF yields, IL-UAE not only increased the TBF content of the extracts about two to three times, but also shortened, observably, the extraction time, and reduced the consumption of solvent. Besides, extraction times of the two methods were basically the same, and the TBF content of IL-UAE was increased nearly two times compared to ethanol-UAE. Through a comparison of the effects of the five extraction methods, it was inferred that IL-UAE had significant potential value to be a quick, effective and environment friendly extract method for TBFs from *S. helvetica*.

### 2.6. Antioxidant of Total Biflavonoid Extract

In order to look for natural antioxidants, the antioxidant activities of total biflavonoid extract (TBFE) from *S. helvetica* were evaluated at different concentrations of each sample (30–150 µg/mL) on the basis of four oxidation determination methods, which were all common antioxidant assays of natural products. As shown in Figure 8A,B, TBFE could obviously inhibit the generation of ABTS^+^ and DPPH^+^ with the increase of concentration of all samples. At 150 µg/mL of each sample, the ABTS^+^ and DPPH^+^ maximum inhibition percentage of TBFE were 89.86% and 81.79%, respectively. However, IC_50_ values of ABTS^+^ and DPPH^+^ scavenging of TBFE (58.63 and 62.69 µg/mL, respectively) were higher than those of quercetin (13.44 and 15.12 µg/mL, respectively) and ascorbic acid (21.19 and 28.76 µg/mL, respectively).

The principle of reducing capacity made Fe^3+^ of potassium ferricyanide change to Fe^2+^ by reduction. Fe^2+^ reacted with ferric trichloride and produced Prussian blue with a maximum absorbance at 700 nm. Based on the principle, if the absorbance measured of antioxygen was stronger, its reducing capacity was higher [39,40]. Figure 9C showed clearly that the reducing power of TBFE, quercetin (QUE), and ascorbic acid (Vc) increased markedly with the variation of sample concentration. According to the methods adopted by Wang and Li [41], the chelation power of ferrous principle used Fe^2+^ to accelerate the transition of the active group, causing oxidative damage. Figure 9D illustrated that two samples had an obvious chelation ability at high concentration, and IC_50_ values of TBFE and EDTA were 49.65 and 44.23 µg/mL, respectively. Moreover, the results were in line with the Fe^3+^ reducing power data mentioned above.

On the basis of the above analysis, the results indicated TBFE from *S. helvetica* had a potential antioxidant effect, which might be related to the hydroxyl groups in biflavonoids [42]. Ji and colleagues [24] considered that the antioxidant effect of flavonoid compounds might mainly rely on the amounts and position of the phenolic hydroxyls at their aromatic ring.

## 3. Material and Methods

### 3.1. Materials

The herb was collected at Simianshan (Zunyi, China) on 09/05/2018, the mud washed off, and dried, naturally, at room temperature. All ionic liquids were purchased from Zhongke Kaite Technology and Trade Co. Ltd. (Lanzhou, China) (see Table 3). Amentoflavone (AME), hinokiflavone (HIN), ginkgetin (GIN), and heveaflavone (HEV) standards (>98% purity) were purchased from Chengdu Ruifensi Biotechnology Co. Ltd. (Chengdu, China). 2, 2-di(4-tert-octylphenyl)-1-picrylhydrazyl (DPPH), 2, 2′-azinobis-(3-ethyl-benzthiaz-oline-6-sulphonate) (ABTS), 3-(2-pyridyl)-5,6-bis (4-sulfo- phenyl)-1,2,4-triazine disodium salt (ferrozine), ethylene diamine tetraacetic acid (EDTA), quercetin, and ascorbic acid were purchased from Tongtian Biotechnology Co. Ltd. (Beijing, China). HPLC methanol and acetonitrile were purchased from Kelon Chemical Reagent Factory (Chengdu, China). All other analytical grade reagents were purchased from Shuangjv Chemical Reagent Factory (Zunyi, China).

### 3.2. Samples Preparation

The raw material of *S. helvetica* was smashed at an impact mill (800Y; Sanmax hardware product Co. Ltd., Xuzhou, China). According to a previous report [43], particle size was controlled through the 100 mesh sieve (17.664–70.425 μm, see Figure 9), which was measured by a LS13320 laser particle diameter analyzer (Runzhi technology Co. Ltd., Jinan, Shandong, China).

### 3.3. IL-Ultrasonic-Assisted Extraction

Under the preestablished ultrasonic power, IL concentration, ratio of liquid to raw material, extraction temperature, and extraction time, the powders of medicinal materials were mingled with a preset amount of extractant. The mixture was poured into the flask and reacted in KQ-500DV ultrasonic extraction apparatus (Kun Shan Ultrasonic Instruments Co. Ltd., Jiangsu, China). The reaction liquid was filtered and centrifuged at 4000 rpm at 5 °C in a MINI-10K high speed centrifuge (Youning Instrument Inc., Hangzhou, China). After concentration and vacuum drying of the filtrate, total biflavonoid extract (TBFE) was acquired, and stored at 5 °C until high performance liquid chromatography (HPLC) analysis.

### 3.4. Determination of Total Biflavonoid Content

TBF content was counted as the summation of the yields of major four biflavonoids, i.e., AME, HIN, GIN, and HEV, which were simultaneously determined through HPLC. A HPLC system was constituted by an Agilent automatic sample handling system series equipped with G1311C-1260 Bin pump, G1329B-1260 automatic column temperature control box and G1315C-1260-detector (Agilent, Santa Clara, CA, USA), and employed to measure total biflavonoids (TBF) in *S. helvetica* extract. Chromatographic separation was performed on an ACE Excel C18 reversed-phase column (5 μm, 250 × 4.6 mm, Phenomenex Technologies, Torrance, CA, USA). Mobile phase was A (acetonitrile) and B (0.1% formic acid solution) using gradient elution (0–10 min, 40–50% (A); 10–20 min, 50–60% (A); 20–30 min, 60–70% (A); 30–40 min, 70–80% (A); 40–50 min, 80–90% (A); 50–60 min, 90–100% (A)). Other chromatographic parameters were 1.0 mL/min flow rate, 20 μL injection volume, 25 °C column temperature, and 330 nm measurement wavelength. The calibration curves for four biflavonoids were YA = 8110.9X − 73.249 (*r* = 0.9996) for AME (0.17–0.61 μg/μL), YB = 2741.6X + 1.0286 (*r* = 0.9995) for HIN (0.16–0.43 μg/μL), YC = 3795.3X + 16.19 (*r* = 0.9998) for GIN (0.11–0.38 μg/μL) and YD = 12280X − 1.9048 (*r* = 0.9997) for HEV (0.10–0.35 μg/μL). Each peak of HPLC profile of TBF extracts was identified by retention times of four standards, i.e., AME, HIN, GIN, HEV, which were 8.5, 15.7, 20.8, and 32.3 min, respectively (Figure 10), and total biflavonoid content of the extract was obtained according to the following formula:TBF content (mg/g) = ((AME amount + GIN amount + HIN amount + HEV amount)/(Initial sample amount)) × 100(2)

### 3.5. Experimental Design

#### 3.5.1. Single-Factor Experiments

To evaluate the test effect on TBF content of the extract from *S. helvetica* by IL-UAE, ultrasonic power (150, 200, 250, 300, 350, 400 W), solvent/material ratio (l:6, 1:8, 1:10, l:12, 1:14, l:16 g/mL), ultrasound time (10, 20, 30, 40, 50, 60 min), IL concentration (0.2, 0.4, 0.6, 0.8, 1.0, 1.2 mmol/L) and ultrasound temperature (20,30, 40, 50, 60, 70 °C) were analyzed as single-factor variates.

#### 3.5.2. RSM Experiments

RSM combined with IL-UAE was often used to analyze and optimize the process parameters in the extraction of natural medicine [44]. In accordance with introduced single-factor experiments, we selected three significant craft parameters, namely, extraction temperature (X_1_), ratio of liquid to raw material (X_2_), and IL concentration (X_3_), as the independent variables (see Table 4). A quadratic polynomial model was used to evaluate the relationship between the response (TBF yields is Y) and three variables (see Table 5), which can be expressed as the following equation:Y = β_0_ + β_1_X_1_ + β_2_X_2_ + β_3_X_3_ + β_11_X_1_^2^ + β_22_X_2_^2^ + β_33_X_3_^2^ + β_12_X_1_X_2_ + β_13_X_1_X_3_ + β_23_X_2_X_3_(3)

### 3.6. Recovery of TBF and IL

The obtained *S. helvetica* extract was diluted 10-fold with deionized water. Then, some insoluble or slightly soluble organic reagents were tested for the extraction of TBF from IL and mobile phase ratio (Vextract: Vorganic reagent) = 1:3. The recovery of TBF was determined according to the above HPLC method in Section 3.4. The recovery of IL was measured by HPLC-DAD. Chromatographic separation condition of IL was performed on a YMC PACK C18 reversed-phase column (5 μm, 250 × 4.6 mm, YMC Technologies, Shimogyo-ku, Kyoto, Japan) using methanol–water (10:90 (*v*/*v*)) as the mobile phase. Other chromatographic parameters were 0.6 mL/min flow rate, 20 μL injection volume, 25 °C column temperature, and 210 nm measurement wavelength. The calibration curves for IL were Y = 8110.9X − 73.249 (*r* = 0.9996) (0.17–0.61 μg/μL) and the retention time of IL was 8.5 min (Figure 11).

### 3.7. Conventional Reference Extraction Methods

#### 3.7.1. Ethanol-Based UAE

According to the method of Wang et al. [45], the powders (15.0 g) were mixed with 190 mL of 95% ethanol in a round bottom flask (250 mL). Afterwards, the mixture was placed in the ultrasonic device, and extracted with 250 W ultrasonic power at 47 °C for 40 min. The reaction liquid was filtered, concentrated (4000 rpm, 5 °C), dried under vacuum, and stored at 4 °C until analysis.

#### 3.7.2. Heat-Reflux Extraction

According to Wang et al. [46] with slight adjustments, the samples (15.0 g) were extracted with 180 mL of 95% ethanol for 120 min in a heat-reflux apparatus (Kelon Glassware Factory, Chengdu, China). After extraction, the extracts were filtered, concentrated (4000 rpm, 5 °C), dried under vacuum, and stored at 4 °C until use.

#### 3.7.3. Soxhelt Extraction

According to the method of Zhao et al. [47], the samples (15.0 g) were kept on a Whatman filter paper in a Soxhlet extractor (Kelon Glassware Factory) and extracted with 250 mL of 95% ethanol for 120 min. Next, the extracts were filtered, concentrated (4000 rpm, 5 °C), dried under vacuum, and stored at 4 °C prior to HPLC analysis.

#### 3.7.4. Percolation Extraction

According to the method of Pezoti et al. [48], the powders (15.0 g) were mixed well with 20 mL of 95% ethanol, placed in a percolator, and soaked for 24 h. Afterwards, 500 mL of effluent fraction was collected with 95% ethanol, and the flow rate was 3 mL/min. The samples collected was filtered, concentrated (4000 rpm, 5 °C), dried under vacuum, and stored at 4 °C until analysis.

### 3.8. Evaluation of Antioxidant Activity

#### 3.8.1. DPPH Radical Scavenging Assay

DPPH radical scavenging ability was determined according to the method [49] with slight modifications. After different concentrations of TBFE (30.0, 60.0, 90.0, 120.0, 150.0 μg/mL) were prepared with ethanol, 0.7 mL of each sample and 2.2 mL of DPPH free radical (0.2 mmol/L) using 95% ethanol preparation were completely mingled, reacted rapidly, and incubated for 60 min at room temperature under a darkened and closed condition, and then the absorbance of the reactant was employed at 517 nm in a 722G ultraviolet spectrophotometer (Shanghai Jingke Industrial Co., Ltd., Shanghai, China). Quercetin and vitamin C (30–150 μg/mL) were adopted as positive controls. The antioxidant capacity of three samples was reflected as an inhibition percent rate of DPPH^+^, and calculated on the grounds of the following formula:*I* (%) = (1 − *A_sample_* /*A_control_*) × 100%(4)where *A_control_* was the absorbance of the control (without TBFE and standards), *A_sample_* was the absorbance of the measured sample, and I was the inhibition percentage rate of the sample. The antioxidant activity was expressed as the effective concentration of the sample providing 50% inhibition rate (IC_50_), and IC_50_ value was calculated by the equation of linear regression.

#### 3.8.2. ABTS Radical Scavenging Assay

ABTS^+^ scavenging ability was evaluated according to the antioxidant method of Chung et al. [50] with some modifications. A 6 mmol/L ABTS solution in 95% ethanol and 120 mmol/L of potassium persulfate water solution were fully mingled, incubated overnight under the darkened condition to produce ABTS^+^. After 0.5 mL of each sample was added to 2.0 mL of ABTS^+^, the mixture was reacted quickly for 10 min at normal temperature. The absorbance of the sample was measured at 734 nm in a 722G ultraviolet spectrophotometer. The IC_50_ value of three samples was calculated by linear regression analysis between the sample of different concentrations and the corresponding inhibition percentage. Quercetin and vitamin C were adopted as reference compounds.

#### 3.8.3. Ferric Ion Reducing Assay

The reducing effect of TBFE was evaluated on the grounds of the method of Oztaskin [51] with slight adjustment. After 1.2 mL of the sample, 1.0 mL of 1% potassium ferricyanide and 1.0 mL of phosphate buffer solution (0.2 mol/L, pH 6.6) were mixed well and reacted completely for 20 min at 50 °C, and the solution was added to 1.0 mL of 10% trichloroacetic acid, mingled fully, and centrifuged. Afterwards, 2.5 mL of the supernatant and 0.5 mL of 0.1% ferric trichloride solution were accurately weighed, blended well and incubated for 10 min at room temperature. Furthermore, the absorbance of the sample was tested at 700 nm in 722G ultraviolet spectrophotometer. Quercetin and vitamin C were adopted as positive controls.

#### 3.8.4. Chelation of Ferrous Ions Assay

The Fe^2+^ chelation of TBFE from *S. helvetica* was evaluated on the grounds of the method of Hentati [52], with some changes. After each sample (0.1 mL) was mixed well with 0.7 mL of ferrous chloride (0.2 mmol/L), the mixture were reacted with 0.6 mL of ferrozine (5 mmol/L) for 10 min at normal temperature resulting in the generation of a purple complex. The absorbance of the sample was tested at 562 nm in 722G ultraviolet spectrophotometer. The IC_50_ value of the two samples was calculated by linear regression analysis between the sample of different concentrations and the corresponding inhibition percentage. EDTA (30–150 μg/mL) was adopted as a positive control.

### 3.9. Statistical Analysis

Design Expert 8.0 (DE, Stat-Ease, Inc., Minneapolis, MN, USA) was adopted for designing and optimizing the process, and obtaining the response models by RSM technology. The analysis of variance (ANOVA) was used to analyze the RSM results and predict the values of TBF. The data were statistically significant at the level of *p* < 0.05 by Duncan’s multiple range test on SPSS 19.0 (Statistical Product and Service Solutions Package for the Social Sciences, IBM, New York, NY, USA). After all the assays were repeated in triplicate, the mean and standard deviation of experimental data were calculated.

## 4. Conclusions

The purpose of this study was to establish a high efficiency, low consumption, environmentally friendly, and novel extraction method. In this test, IL-UAE was successfully applied for the extraction of biflavonoids from *S. helvetica* for the first time, according to the optimization of single-factor test and RSM. Among the three dependent variates in RSM, IL concentration for TBF yields was the most notable factor, with its *p* < 0.001. The optimum parameters for TBFE were ultrasonic power 250 W, IL concentration 0.8 mmol/L, extraction time 40 min, solid–liquid ratio 1:12.7 g/mL, and extraction temperature 47 °C. The yield of TBF was 18.69 mg/g under the best process conditions. Moreover, the recommended IL-UAE was more effective than ethanol-UAE, SE, PE, and HRE methods in higher TBF content, less solvent, and shorter time. In addition, TBFE had shown suitable antioxidant activity in four antioxidant assays. Thus, this study confirmed that *S. helvetica* could be a good source of natural antioxidants and IL-UAE technique was an effective method for extracting TBF from the *Selaginella* plant.

## Figures and Tables

**Figure 1 molecules-23-03284-f001:**
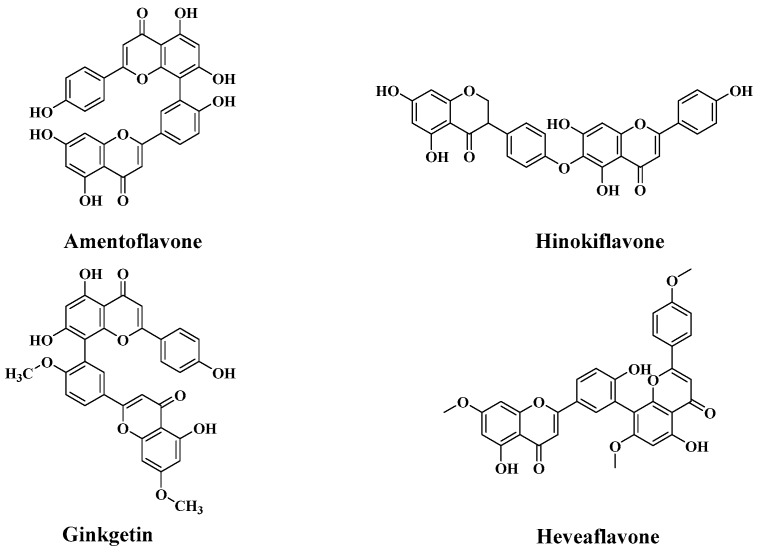
Chemical structures of four biflavonoids from *Selaginella helvetica*.

**Figure 2 molecules-23-03284-f002:**
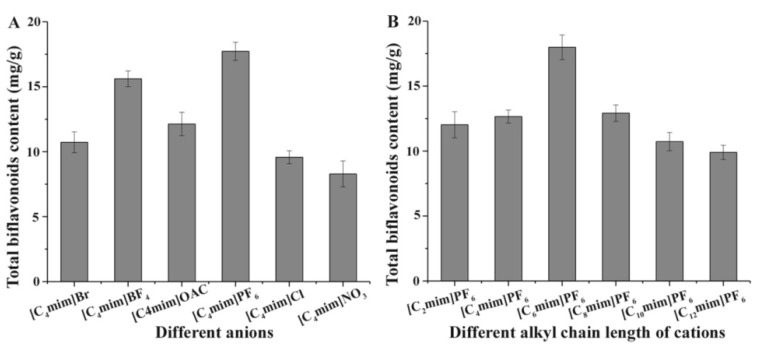
Effect of the (**A**) ionic liquid (IL) anions and (**B**) IL cations for total biflavonoid content from *Selaginella helvetica*.

**Figure 3 molecules-23-03284-f003:**
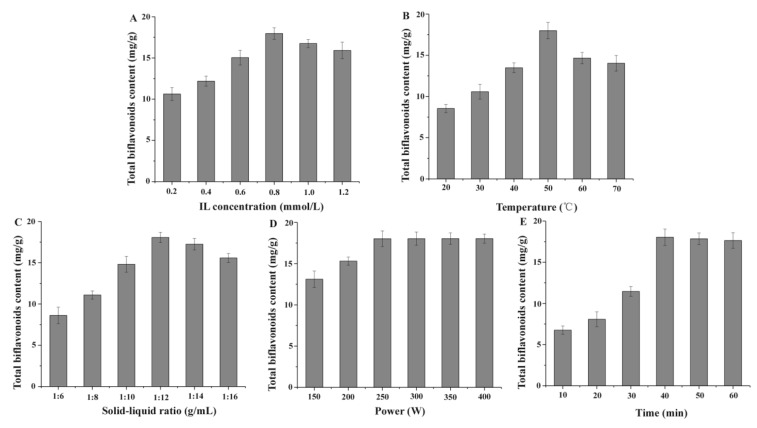
Effects of extraction parameters on total biflavonoid yields of *Selaginella helvetica* extracts. (**A**) IL concentration, (**B**) extraction temperature, (**C**) ratio of solid to liquid, (**D**) ultrasound power, and (**E**) extraction time.

**Figure 4 molecules-23-03284-f004:**
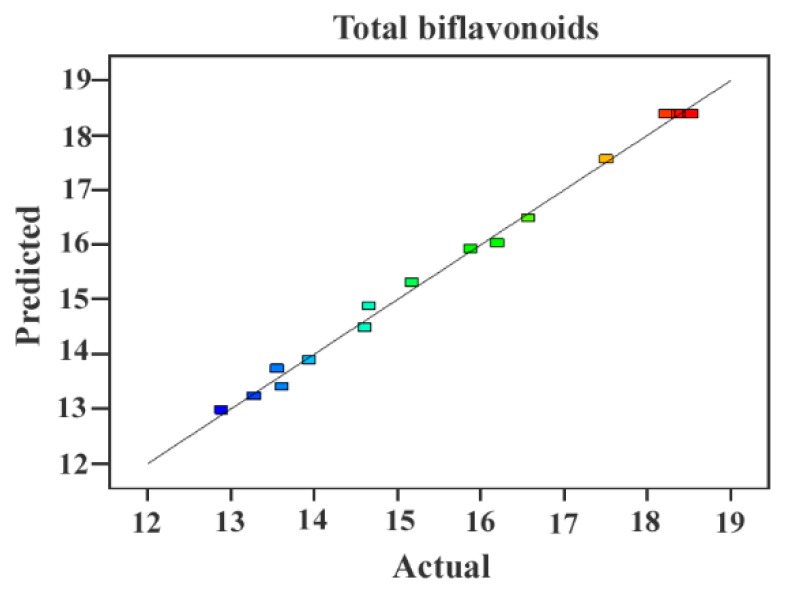
Correlation between the predicted and actual values for total biflavonoid content. Color points from blue to red show values of biflavonoids changing in the range of 12–19 mg/g.

**Figure 5 molecules-23-03284-f005:**
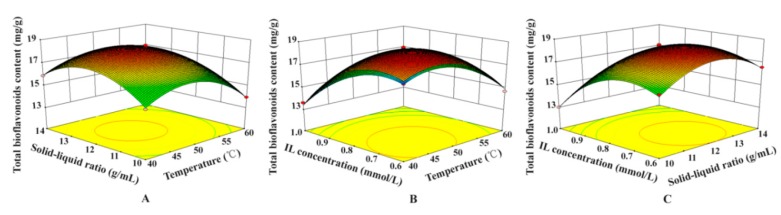
Response surface plots showing interaction effects of solid–liquid ratio and extraction temperature (**A**); extraction temperature and IL concentration (**B**); solid–liquid ratio and IL concentration (**C**) on total biflavonoid yields.

**Figure 6 molecules-23-03284-f006:**
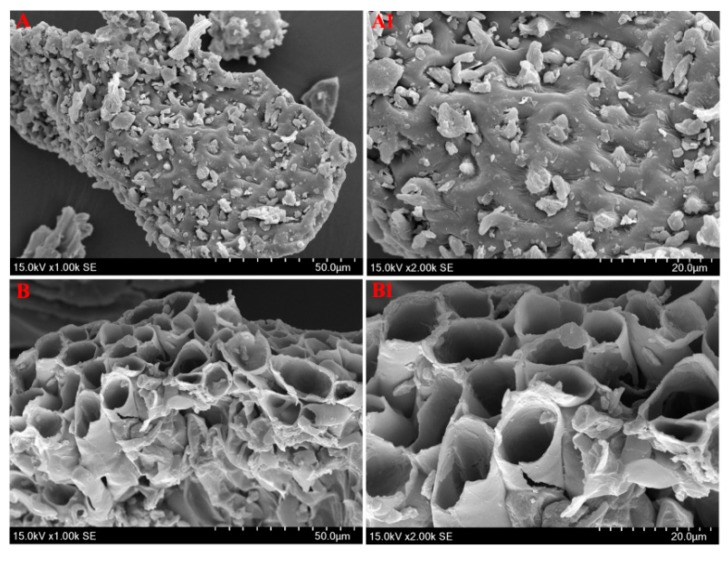
SEM graphics of total biflavonoid extract (TBFE) from *S. helvetica*. Raw materials (**A**,**A1**) and treated samples by IL-UAE (**B**,**B1**) were observed under 1000 and 2000 magnification, respectively.

**Figure 7 molecules-23-03284-f007:**
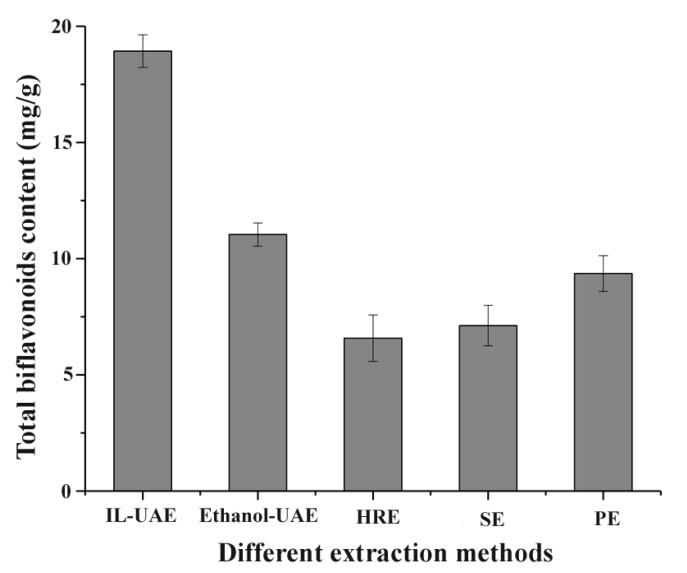
Effect of five extraction methods for biflavonoid content from *Selaginella helvetica*.

**Figure 8 molecules-23-03284-f008:**
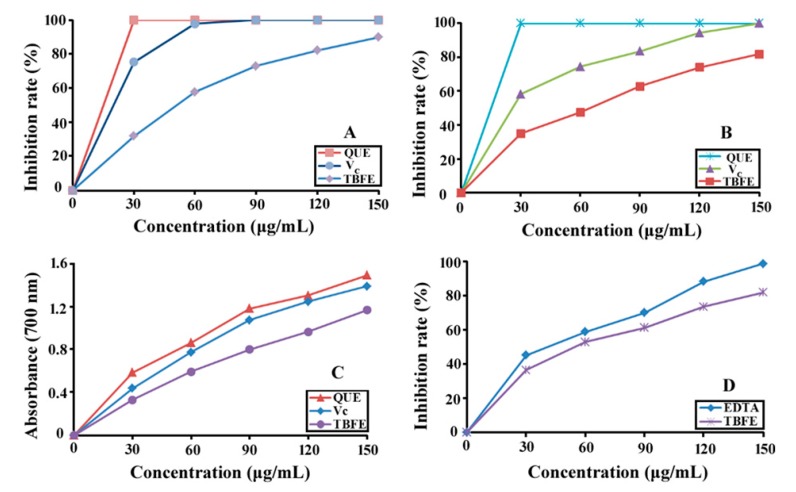
Antioxidant effects of TBFE, quercetin (QUE), and ascorbic acid (Vc). (**A**) ABTS radical scavenging activity; (**B**) DPPH radical scavenging activity; (**C**) reducing ability; (**D**) ferric chelation ability.

**Figure 9 molecules-23-03284-f009:**
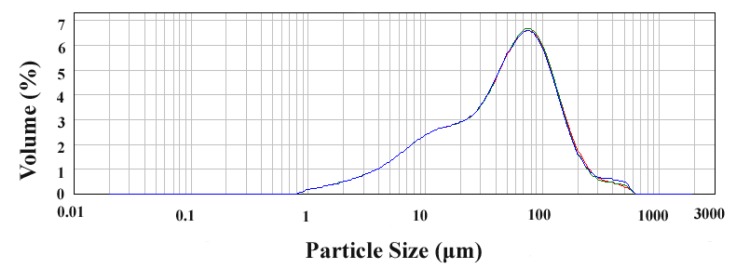
Particle diameter distribution of *S. helvetica*.

**Figure 10 molecules-23-03284-f010:**
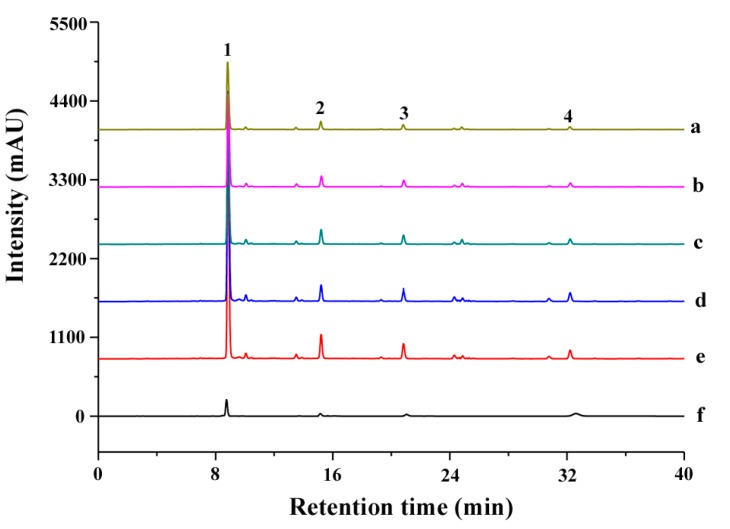
The typical HPLC chromatographic profile of mixed standards and total biflavonoid extract acquired in five extraction methods. (**a**) Heat-reflux extraction; (**b**) Soxhlet extraction; (**c**) percolation extraction; (**d**) ethanol-UAE; (**e**) IL-UAE; (**f**) mixed standards. The peaks marked with 1, 2, 3, and 4, which were AME, HIN, GIN, and HEV, respectively.

**Figure 11 molecules-23-03284-f011:**
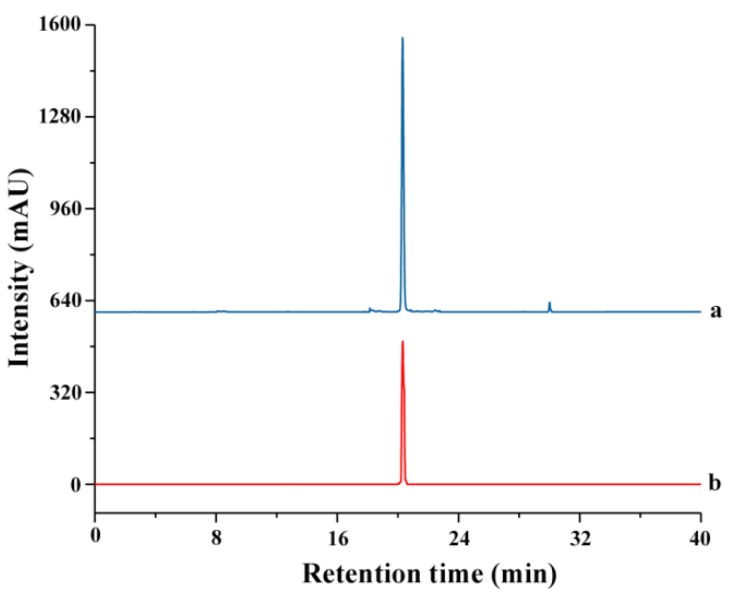
HPLC-DAD chromatograms of recovery sample from *S. helvetica* (**a**) and [C_6_mim]PF_6_ standard (**b**).

**Table 1 molecules-23-03284-t001:** ANOVA for the fitted secondary order curve model for total biflavonoid content.

Source	Sum of Squares	df	*F* Value	*p* Value	*R* ^2^	*R*^2^ (Adj)	Significant
Model	67.92	9	180.95	<0.0001	0.9957	0.9902	significant
X_1_	4.06	1	97.38	0.0015			
X_2_	0.73	1	17.41	0.0042			
X_3_	16.79	1	402.60	<0.0001			
X_1_X_2_	2.250 × 10^−4^	1	5.395 × 10^−3^	0.9435			
X_1_X_3_	1.58	1	37.76	0.0005			
X_2_X_3_	0.023	1	0.54	0.4865			
X_1_^2^	13.04	1	312.54	<0.0001			
X_2_^2^	12.49	1	299.36	<0.0001			
X_3_^2^	14.52	1	348.14	<0.0001			
Residual	0.29	7					
Lack of Fit	0.21	3	3.37	0.1355			not significant
Pure Error	0.083	4					
Cor Total	68.21	16					

**Table 2 molecules-23-03284-t002:** Results of recovery of TBFs and IL with different organic solvents (%) (*n* = 3).

	Ethyl Acetate	Chloroform	*n*-Butanol	Dichloromethane	Ether	*n*-Hexane
TBFs	96.23 ± 0.53	38.22 ± 0.31	87.56 ± 0.28	50.75 ± 0.37	73.69 ± 0.34	15.76 ± 0.49
IL	97.67 ± 0.44	94.59 ± 0.45	97.21 ± 0.39	98.46 ± 0.25	95.17 ± 0.52	96.29 ± 0.38

**Table 3 molecules-23-03284-t003:** Chemical structures of ionic liquids.

Ionic Liquids	Cations	Anions
[C_4_mim]Cl (1-butyl-3-methylimidazolium-chloride)	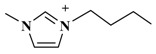	Cl^−^
[C_4_mim]NO_3_ (1-butyl-3-methylimidazolium-nitrate)	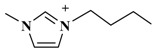	NO_3_^−^
[C_4_mim]PF_6_ (1-butyl-3-methylimidazolium-hexafluorophosphate)	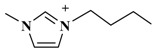	PF_6_^−^
[C_4_mim]Br (1-butyl-3-methylimidazolium-bromine)	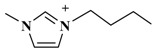	Br^−^
[C_4_mim]BF_4_ (1-butyl-3-methylimidazolium-tetrafluoroborate)	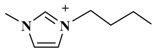	BF_4_^−^
[C_4_mim]CH_3_COO (1-butyl-3-methyli-midazolium-acetate)	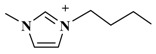	CH_3_COO^−^
[C_2_mim]PF_6_ (1-ethyl-3-methylimidazolium-hexafluorophosphate)	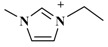	PF_6_^−^
[C_6_mim]PF_6_ (1-hexyl-3-methylimidazolium-hexafluorophosphate)	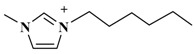	PF_6_^−^
[C_8_mim]PF_6_ (1-octyl-3-methylimida-zolium-hexafluorophosphate)	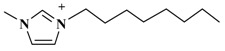	PF_6_^−^
[C_10_mim]PF_6_ (1-decyl-3-methylimidazolium-hexafluorophosphate)	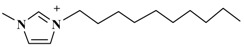	PF_6_^−^
[C_12_mim]PF_6_ (1-dodecyl-3-methylimidazolium-hexafluorophosphate)	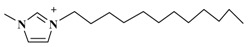	PF_6_^−^

**Table 4 molecules-23-03284-t004:** Levels and factors selected for experiment.

Level		Factors	
	Solid–Liquid Ratio (g/mL)	IL Concentration (mmol/L)	Extract Temperature (°C)
−1	1:10	0.6	40
0	1:12	0.8	50
1	1:14	1.0	60

**Table 5 molecules-23-03284-t005:** The results of varying experiment design.

Run	X_1_	X_2_	X_3_	Y
	Extract Temperature (°C)	Solid–Liquid Ratio (g/mL)	IL Concentration (mmol/L)	Yield of TBFs (mg/g)
1	60	12	0.6	14.66
2	60	10	0.8	13.94
3	60	14	0.8	14.61
4	50	12	0.8	18.22
5	50	12	0.8	18.51
6	50	14	0.6	16.57
7	40	12	1.0	13.62
8	40	10	0.8	15.18
9	50	12	0.8	18.25
10	50	14	1.0	13.56
11	50	12	0.8	18.41
12	50	10	1.0	12.89
13	50	10	0.6	16.20
14	40	12	0.6	17.51
15	60	12	1.0	13.28
16	50	12	0.8	18.53
17	40	14	0.8	15.88

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
