# Peer review of "Ionic Liquid–Ultrasound-Based Extraction of Biflavonoids from Selaginella helvetica and Investigation of Their Antioxidant Activity"

_molecules, 2018, doi:10.3390/molecules23123284_

Reviewer 1 Report

Overall, the authors did a nice job clarifying procedures, making important corrections, and adding meaningful references. In terms of the procedures employed and the data analyses, I am fine with most of the revisions. However, too many language errors remain, including many new errors in the added sentences and paragraphs. These should be corrected prior to publication. There are far too many to list, but I will point out several for the authors’ benefit:

Line 108: “stereo hindrance” is not a correct term. ‘Steric hindrance’ is correct.

Line 122: This line is not logical as written. It should be revised.

Lines 144-146: The sentence does not make sense as written. Perhaps it should read, “Plausible reasons for the low TBF yield could include thermal instability and phenolic oxidation at isolation temperatures >50 oC.”

Lines 154-155: This sentence does not make sense as written.

Line 157: The written phrase is not a complete sentence and is nonsensical.

Line 170: the last clause should read, “thereby increasing the extraction efficiency.”

Line 264: “to aspire to be” is incorrect usage. Perhaps ‘as’ is a suitable replacement.

Line 275: there appears to be an error here, in that the IC50 values are higher, not lower, as stated.

Line 289: “antioxidation on account of its hydroxyl groups contained” makes no sense. This must be clarified. In addition, I do not find “antioxidation” an acceptable term, and it should be changed throughout the document.

Line 378: “filtrated” should be changed to ‘filtered’ as correctly written in other procedures.

Line 397: “0.7 mL of the eluted each sample” is very unclear. Do the authors mean, ”a 0.7 mL portion of each sample was eluted”?

Line 443: “green direction” is not an appropriate adjective. Perhaps “more environmentally friendly” or similar would fit better.

Line 447: “remarkable” is an ambiguous usage; ‘notable’ is more appropriate.

Again, that is only a very partial list of errors and confusing statements. Once the items indicated above as well as the many other grammar errors are fixed, I would favor publication in Molecules.

Author Response

Dear Editors and Reviewer:

We do extremely appreciate your suggestions. Those comments are all valuable and very helpful for revising and improving our paper, as well as the important guiding significance to our researches. We have made careful investigation, revision and improvement for the original manuscript. Main changes in the revised manuscript have been marked as highlighted text in the paper. We hope the revised manuscript will meet your requirement. The following content is our point-by-point responses to your comments and questions.

Looking forward to hearing from you soon, thank you very much!

Yours Sincerely

Authors

 Response to reviewer

1. Line 108: “stereo hindrance” is not a correct term. ‘Steric hindrance’ is correct.

Response: Thank you for this direction. We have changed “stereo hindrance” to “steric hindrance” in the sentence.

2. Line 122: This line is not logical as written. It should be revised.

Response: Many thanks for your warning. We have revised the logical mistakes in the manuscript.

3. Lines 144-146: The sentence does not make sense as written. Perhaps it should read, “Plausible reasons for the low TBF yield could include thermal instability and phenolic oxidation at isolation temperatures >50 °C.”

Response: Thank you for reminding me! According to your suggestion, we have carefully modified the sentence in the manuscript.

4. Lines 154-155: This sentence does not make sense as written.

Response: Many thanks for your advice. We have modified the sentence and made it sense in the manuscript.

5. Line 157: The written phrase is not a complete sentence and is nonsensical.

Response: Thank you for this direction. We have carefully revised the errors in the manuscript.

6. Line 170: the last clause should read, “thereby increasing the extraction efficiency.”

Response: Thanks for your comment! According to your suggestion, we have revised the sentence in the manuscript.

7. Line 264: “to aspire to be” is incorrect usage. Perhaps ‘as’ is a suitable replacement.

Response: We have revised the errors in the manuscript.

8. Line 275: there appears to be an error here, in that the IC50 values are higher, not lower, as stated.

Response: Sorry, the expression is improper. We have revised the error.

9. Line 289: “antioxidation on account of its hydroxyl groups contained” makes no sense. This must be clarified. In addition, I do not find “antioxidation” an acceptable term, and it should be changed throughout the document.

Response: Thanks for the very meaningful suggestion! We have carefully modified the sentence and changed “antioxidation” to “antioxidant” throughout the document.

10. Line 378: “filtrated” should be changed to ‘filtered’ as correctly written in other procedures.

Response: Sorry, the expression is improper. We have modified “filtrated” to “filtered” throughout the manuscript.

11. Line 397: “0.7 mL of the eluted each sample” is very unclear. Do the authors mean, ”a 0.7 mL portion of each sample was eluted”?

Response: Thank you for reminding me! We have changed “0.7 mL of the eluted each sample” to “0.7 mL of each sample” in the sentence.

12. Line 443: “green direction” is not an appropriate adjective. Perhaps “more environmentally friendly” or similar would fit better.

Response: We have changed “green direction” to “being environmentally friendly” in the manuscript.

13. Line 447: “remarkable” is an ambiguous usage; ‘notable’ is more appropriate.

Response: Yes, the expression is improper. We have changed it to “notable” in the sentence.

14. Again, that is only a very partial list of errors and confusing statements. Once the items indicated above as well as the many other grammar errors are fixed, I would favor publication in Molecules.

Response: Thank you for reminding me! Based on the items indicated above, we have carefully checked and revised other grammar errors throughout the manuscript.

Reviewer 2 Report

Thank you very much for the extensive revision and extension of the presented study. All additional information is extremly helpful and benefits the reader very much.

Author Response

Dear Editors and Reviewer:

We do extremely appreciate your suggestions. Those comments are all valuable and very helpful for revising and improving our paper, as well as the important guiding significance to our researches. We have made careful investigation, revision and improvement for the original manuscript. Main changes in the revised manuscript have been marked as highlighted text in the paper. We hope the revised manuscript will meet your requirement. The following content is our point-by-point responses to your comments and questions.

Looking forward to hearing from you soon, thank you very much!

Yours Sincerely

Authors

Response to reviewer

1. Thank you very much for the extensive revision and extension of the presented study. All additional information is extremely helpful and benefits the reader very much.

Response: Thanks very much for your review, which has benefited me a lot. On behalf of my co-authors, we would like to express our great appreciation to editor and reviewers.